# Metric Learning for Adversarial Robustness

**Chengzhi Mao**
Columbia University
cm3797@columbia.edu

**Ziyuan Zhong**
Columbia University
ziyuan.zhong@columbia.edu

**Junfeng Yang**
Columbia University
junfeng@cs.columbia.edu

**Carl Vondrick**
Columbia University
vondrick@cs.columbia.edu

**Baishakhi Ray**
Columbia University
rayb@cs.columbia.edu

## Abstract

Deep networks are well-known to be fragile to adversarial attacks. We conduct an empirical analysis of deep representations under the state-of-the-art attack method called PGD, and find that the attack causes the internal representation to shift closer to the "false" class. Motivated by this observation, we propose to regularize the representation space under attack with metric learning to produce more robust classifiers. By carefully sampling examples for metric learning, our learned representation not only increases robustness, but also detects previously unseen adversarial samples. Quantitative experiments show improvement of robustness accuracy by up to 4% and detection efficiency by up to 6% according to Area Under Curve score over prior work. The code of our work is available at https://github.com/columbia/Metric_Learning_Adversarial_Robustness.

## 1   Introduction

Deep networks achieve impressive accuracy and wide adoption in computer vision [17], speech recognition [14], and natural language processing [21]. Nevertheless, their performance degrades under adversarial attacks, where natural examples are perturbed with human-imperceptible, carefully crafted noises [35, 23, 12, 18]. This degradation raises serious concern — especially when we deploy deep networks to safety and reliability critical applications [29, 43, 41, 20, 36]. Extensive efforts [37, 31, 47, 7, 25, 12, 35, 48] have been made to study and enhance the robustness of deep networks against adversarial attacks, where a defense method called adversarial training achieves the state-of-the-art adversarial robustness [19, 16, 46, 49].

To better understand adversarial attacks, we first conduct an empirical analysis of the latent representations under attack for both defended [19, 16] and undefended image classification models. Following the visualization technique in [28, 30, 33], we investigate what happens to the latent representations as they undergo attack. Our results show that the attack shifts the latent representations of adversarial samples away from their *true* class and closer to the *false* class. The adversarial representations often spread across the false class distribution in such a way that the natural images of the false class become indistinguishable from the adversarial images.

Motivated by this empirical observation, we propose to add an additional constraint to the model using metric learning [15, 32, 44] to produce more robust classifiers. Specifically, we add a triplet loss term on the latent representations of adversarial samples to the original loss function. However, the naïve implementation of triplet loss is not effective because the pairwise distances of a natural sample $x_a$, its adversarial sample $x'_a$, and a randomly selected natural sample of the false class $x_n$ are hugely uneven. Specifically, given considerable data variance in the false class, $x_n$ is often far from the decision boundary where $x'_a$ resides, therefore $x_n$ is too easy as a negative sample. To address this

problem, we sample the negative example for each triplet with the closest example in a mini-batch of training data. In addition, we randomly select another sample $x_p$ in the correct class as the positive example in the triplet data.

Our main contribution is a simple and effective metric learning method, Triplet Loss Adversarial (TLA) training, that leverages triplet loss to produce more robust classifiers. TLA brings near both the natural and adversarial samples of the same class while enlarging the margins between different classes (Sec. 3). It requires no change to the model architecture and thus can improve the robustness on most off-the-shelf deep networks without additional overhead during inference. Evaluation on popular datasets, model architectures, and untargeted, state-of-the-art attacks, including projected gradient descent (PGD), shows that our method classifies adversarial samples more accurately by up to 4% than prior robust training methods [16, 19]; and makes adversarial attack detection [52] more effective by up to 6% according to the Area Under Curve (AUC) score.

## 2  Related Work

The fact that adversarial noise can fool deep networks was first discovered by Szegedy et al. [35], which started the era of adversarial attacks and defenses for deep networks. Goodfellow et al. [12] then proposed an attack — fast gradient sign method (FGSM) and also constructed a defense model by training on the FGSM adversarial examples. More effective attacks including C&W [5], PGD [19], BIM [18], MIM [9], DeepFool [23], and JSMA [27] are proposed to fool deep networks, which further encourage the research for defense methods.

Madry et al. [19] proposed adversarial training (AT) that dynamically trained the model on the generated PGD attacks, achieving the first empirical adversarial robust classifier on CIFAR-10. Since then, AT became the foundation for the state-of-the-art adversarial robust training method and went through widely and densely scrutiny [3], which achieved real robustness without relying on gradient masking [3, 13, 31, 4, 8]. Recently, Adversarial Logit Pairing (ALP) [16] is proposed with an additional loss term that matches the logit feature from a clean image $\mathbf{x}$ and its corresponding adversarial image $\mathbf{x}'$, which further improves the adversarial robustness. However, this method has a distorted loss function and is not scalable to untargeted attack [11, 22]. In contrast to the ALP loss which uses a pair of data, our method introduces an additional negative example in a triplet of data, which achieves more desirable geometric relationships between adversarial examples and clean examples in feature metric space.

Orthogonal to our method, the concurrent feature denoising method [46] achieves the state-of-the-art adversarial robustness on ImageNet. While their method adds extra denoising block in the model, our method requires no change to the model architecture. Another concurrent work, TRADES [49], achieves improved robustness by introducing Kullback-Leibler divergence loss to a pair of data. In addition, unlabeled data [39] and model ensemble [37, 25] have been shown to improve the robustness of the model. Future work can be explored by combining these methods with our proposed TLA regularization for better adversarial robustness.

## 3  Qualitative Analysis of Latent Representations under Adversarial Attack

We begin our investigation by analyzing how the adversarial images are represented by different models. We call the original class of an adversarial image as *true* class and the mis-predicted class of adversarial example as *false* class. Figure 1 shows the visualization of the high dimensional latent representation of sampled CIFAR-10 images with t-SNE [40, 2]. Here, we visualize the penultimate fully connected (FC) layer of four existing models: standard undefended model (UM), model after adversarial training (AT) [19], model after adversarial logit pairing (ALP) [16], and model after our proposed TLA training. Though all the adversarial images belong to the same *true* class, UM separates them into different *false* classes with large margins. The result shows UM is highly non-robust against adversarial attacks because it is very easy to craft an adversarial image that will be mistakenly classified into a different class. With AT and ALP methods, the representations are getting closer together, but one can still discriminate them. Note that, a good robust model will bring the representations of the adversarial images closer to their original *true* class so that it will be difficult to discriminate the adversarial images from the original images. We will leverage this observation to design our approach.

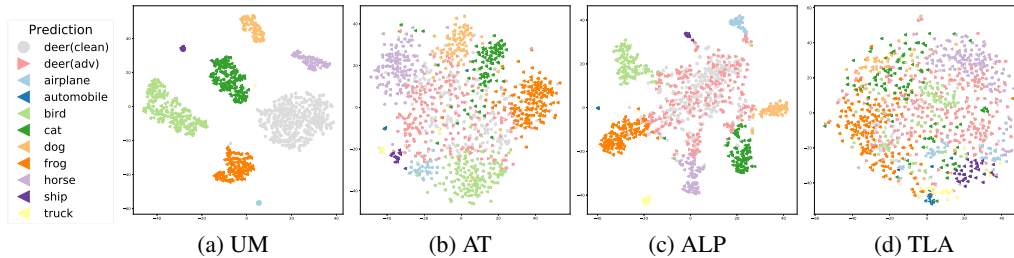

(a) UM        (b) AT        (c) ALP        (d) TLA

Figure 1: **t-SNE Visualization of adversarial images from the same *true* class which are mistakenly classified to different *false* classes.** The figure shows representations of second to last layer of 1000 adversarial examples crafted from 1000 natural (clean) test examples from CIFAR-10 dataset, where the *true* class is "deer." The different colors represent different *false* classes. The gray dots further show 500 randomly sampled natural deer images. Notice that for (a) undefended model (UM), the adversarial attacks clearly separate the images from the same "deer" category into different classes. (b) adversarial training (AT) and (c) adversarial logit pairing (ALP) method still suffer from this problem at a reduced level. In contrast, our proposed ATL (see (d)) clusters together all the examples from the same *true* class, which improves overall robustness.

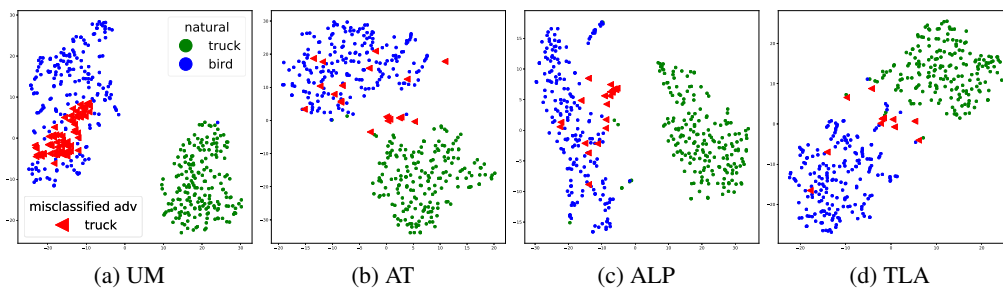

(a) UM        (b) AT        (c) ALP        (d) TLA

Figure 2: **Illustration of the separation margin of adversarial examples from the natural images of the corresponding false class.** We show t-SNE visualization of the second to last layer representation of test data from two different classes across four models. The blue and green dots are 200 randomly sampled natural images from "bird" and "truck" classes respectively. The red triangles denote adversarial (adv) truck images but mispredicted as "bird." Notice that for (a) UM, the adversarial examples are moved to the center of the false class, making it hard to separate from them. (b) AT and (c) ALP achieve some robustness by separating adversarial and natural images, but they are still close to each other. Plot (d) shows our proposed TLA training promotes the mispredicted adversarial examples to lie on the edge of the natural images false class and can still be separated, which improves the robustness.

In Figure 2, we further analyze how the representation of images of one class is attacked into the neighborhood of another class. The green and blue dots are the natural images of trucks and birds, respectively. The red triangles are the adversarial images of trucks mispredicted as birds. For UM model (Figure 2a), all the adversarial attacks successfully get into the center of the false class. The AT and ALP models achieve some robustness by separating some adversarial images from natural images, but most adversarial images are still inside the false class. A good robust model should promote the representations of adversarial examples away from the false class, as shown in Figure 2d. Such separation not only improves the adversarial classification accuracy but also helps to reject the mispredicted adversarial attacks, because the mispredicted adversaries tend to lie on edge.

Based on these two observations, we build a new approach that ensures adversarial representations will be (i) closer to the natural image representations of their true classes, and (ii) farther from the natural image representations of the corresponding false classes.

# 4 Approach

Inspired by the adversarial feature space analysis, we add an additional constraint to the model using metric learning. Our motivation is that the triplet loss function will pull all the images of one class, both natural and adversarial, closer while pushing the images of other classes far apart. Thus, an image and its adversarial counterpart should be on the same manifold, while all the members of the false class should be forced to be separated by a large margin.

**Notations.** For an image classification task, let $M$ be the number of classes to predict, and $N$ be the number of training examples. We formulate the deep network classifier as $F_{\boldsymbol{\theta}}(\mathbf{x}) \in \mathbb{R}^M$ as a probability distribution, where $\mathbf{x}$ is the input variable, $\mathbf{y}$ is the output ground-truth, and $\boldsymbol{\theta}$ is the network's parameters to learn (we simply use $F(\mathbf{x})$ most of time); $\mathcal{L}(F(\mathbf{x}), y)$ is the loss function.

Assume that an adversary is capable of launching adversarial attacks bounded by $p$-norm, i.e., the adversary can perturb the input pixel by $\epsilon$ bounded by $L_p, p = 0, 2, \infty$, let $\mathbf{I}(\mathbf{x}, \epsilon)$ denote the $L_p$ ball centered at $\mathbf{x}$ with radius $\epsilon$. We focus on the study of *untargeted* attack, i.e., the objective is to generate $\mathbf{x}' \in \mathbf{I}(\mathbf{x}, \epsilon)$ such that $F(\mathbf{x}') \neq F(\mathbf{x})$.

**Triplet Loss.** Triplet loss is a widely used strategy for metric learning. It trains on a triplet input $\left\{ \left( \mathbf{x}_a^{(i)}, \mathbf{x}_p^{(i)}, \mathbf{x}_n^{(i)} \right) \right\}$, where the elements in the positive pair $\left( \mathbf{x}_a^{(i)}, \mathbf{x}_p^{(i)} \right)$ are clean images from the same class and the elements in the negative pair $\left( \mathbf{x}_a^{(i)}, \mathbf{x}_n^{(i)} \right)$ are from different classes [32, 15]. $\mathbf{x}_p^{(i)}$, $\mathbf{x}_a^{(i)}$, and $\mathbf{x}_n^{(i)}$ are referred as *positive*, *anchor*, and *negative* examples of the triplet loss. The embeddings are optimized such that examples of the same class are pulled together and the examples of different classes are pushed apart by some margin [34]. The standard triplet loss for clean images is as follows:

$$\sum_i^N \mathcal{L}_{trip}(\mathbf{x}_a^{(i)}, \mathbf{x}_p^{(i)}, \mathbf{x}_n^{(i)}) = \sum_i^N [D(h(\mathbf{x}_a^{(i)}), h(\mathbf{x}_p^{(i)})) - D(h(\mathbf{x}_a^{(i)}), h(\mathbf{x}_n^{(i)})) + \alpha]_+$$

where, $h(\mathbf{x})$ maps from the input $\mathbf{x}$ to the embedded layer, $\alpha \in \mathbb{R}^+$ is a hyper-parameter for margin and $D(h(\mathbf{x}_i), h(\mathbf{x}_j))$ denotes the distance between $\mathbf{x}_i$ and $\mathbf{x}_j$ in the embedded representation space. In this paper, we define the embedding distance between two examples using the angular distance [42]: $D(h(\mathbf{x}_a^{(i)}), h(\mathbf{x}_{p,n}^{(j)})) = 1 - \frac{|h(\mathbf{x}_a^{(i)}) \cdot h(\mathbf{x}_{p,n}^{(j)})|}{||h(\mathbf{x}_a^{(i)})||_2 ||h(\mathbf{x}_{p,n}^{(j)})||_2}$, where we choose to encode the information in the angular metric space.

**Metric Learning for Adversarial Robustness.** We add triplet loss to the penultimate layer's representation. Different from standard triplet loss where all the elements in the triplet loss term are clean images [32, 50], at least one element in the triplet loss under our setting will be an adversarial image. Note that generating adversarial examples is more computational intensive compared with just taking the clean images. For efficiency, we only generate one adversarial perturbed image for each triplet data, using the same method introduced by Madry et al. [19]. Specifically, given a clean image $\mathbf{x}^{(i)}$, we generate the adversarial image $\mathbf{x}'^{(i)}$ based on $\nabla_{\mathbf{x}} \mathcal{L}(F(\mathbf{x}), y)$ (standard loss without the triplet loss) with PGD method. We do not add the triplet loss term into the loss of adversarial example generation due to its inefficiency.

The other elements in the triplet data are clean images. We forward the triplet data in parallel through the model and jointly optimize the cross-entropy loss and the triplet loss, which enables the model to capture the stable metric space representation (triplet loss) with semantic meaning (cross-entropy loss). The total loss function is formulated as follows:

$$\mathcal{L}_{all} = \sum_i^N \mathcal{L}_{ce}(f(\mathbf{x}_a'^{(i)}), y^{(i)}) + \lambda_1 \mathcal{L}_{trip}(h(\mathbf{x}_a'^{(i)})), h(\mathbf{x}_p^{(i)}), h(\mathbf{x}_n^{(i)})) + \lambda_2 \mathcal{L}_{norm} \tag{1}$$

$$\mathcal{L}_{norm} = ||h(\mathbf{x}_a'^{(i)})||_2 + ||h(\mathbf{x}_p^{(i)})||_2 + ||h(\mathbf{x}_n^{(i)})||_2$$

where $\lambda_1$ is a positive coefficient trading off the two losses; $\mathbf{x}_a'^{(i)}$ (anchor example) is an adversarial counterpart based on $\mathbf{x}_a^{(i)}$; $\mathbf{x}_p^{(i)}$ (positive example) is a clean image from the same class of $\mathbf{x}_a^{(i)}$; $\mathbf{x}_n^{(i)}$ (negative example) is a clean image from a different class; $\lambda_2$ is the weight for the feature norm decay term, which is also applied in [32] to reduce the $L_2$ norm of the feature.

Notice that, besides the TLA set-up in equation 1, an adversarial perturbed image can be the positive example, and a clean image can be the anchor example (i.e., switch the anchor and the positive), where we refer it as TLA-SA (Sec 5). We choose the adversarial example as the anchor for TLA according to the experimental result. Intuitively, the adversarial image is picked as the anchor because it tends to be closer to the decision boundary between the "true" class and the "false" class. As an anchor, the adversarial example is considered in both the positive pair and the negative pair, which gives more-useful gradients for the optimization. The modified triplet loss for adversarial robustness is shown in Figure 3.

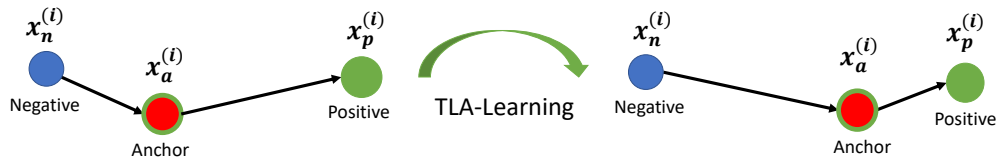

Figure 3: Illustration of the triplet loss for adversarial robustness (TLA). The red circle is an adversarial example, while the green and the blue circles are clean examples. The anchor and positive belong to the same class. The negative (blue), from a different class, is the closest image to the anchor (red) in feature space. TLA learns to pull the *anchor* and *positive* from the true class closer, and push the *negative* of false classes apart.

**Negative Sample Selection.** In addition to the anchor selection, the selection of the negative example is crucial for the training process, because most of the negative examples are easy examples that already satisfy the margin constraint of pairwise distance and thus contribute useless gradients [32, 10]. Using the representation angular distance we predefine, we select negative samples as the nearest images to the anchor from a false class. As a result, our model is able to learn to enlarge the boundary between the adversarial samples and their closest negative samples from the other classes.

Unfortunately, finding the closest negative samples from the entire training set is computationally intensive. Besides, using very hard negative examples have been found to decrease the network's convergence speed [32] significantly. Instead, we use a semi-hard negative example, where we select the closest sample in a mini-batch. We demonstrate the advantage of this sampling strategy by comparing it with the random sampling (TLA-RN). The results are shown in Sec 5. Other strategies of sampling negative samples such as DAML [10] could also be applied here, which uses an adversarial generator to exploit hard negative examples from easy ones.

**Implementation Details.** We apply our proposed triplet loss on the embedding of the penultimate layer of the neural network for classification tasks. Since the following transformation only consists of a linear layer and a softmax layer, small fluctuation to this embedding only brings monotonous adjustment to the output controlled by some tractable Lipschitz constant [7, 24]. We do not apply triplet loss on the logit layer but on the penultimate layer, because the higher dimensional penultimate layer tends to preserve more information. We also construct two triplet loss terms on CIFAR-10 and Tiny ImageNet, adding another positive example while reusing the anchor and negative example, which achieves better performance [34, 6]. The details of the algorithm are introduced in the appendix.

## 5   Experiments

**Experimental Setting.** We validate our method on different model architectures across three popular datasets: MNIST, CIFAR-10, and Tiny-ImageNet. We compare the performance of our models with the following baselines: **Undefended Model (UM)** refers to the standard training without adversarial samples, **Adversarial Training (AT)** refers to the min-max optimization method proposed in [19], **Adversarial Logit Pairing (ALP)** refers to the logit matching method which is currently the state-of-the-art [16]. We use **TLA** to denote the triplet loss adversarial training mentioned in Section 4. To further evaluate our design choice, we study two variants of TLA: **Random Negative (TLA-RN)**, which refers to our proposed triplet loss training method with a randomly sampled negative example, and **Switch Anchor (TLA-SA)**, which sets the anchor to be natural example and the positive to be adversarial example (i.e., switching the anchor and the positive of our proposed method).

We conduct all of our experiments using TensorFlow v1.13 [1] on a single Tesla V100 GPU with a memory of 16GB. We adopt the untargeted adversarial attacks during all of our training processes, and evaluate the models with both white-box and black-box *untargeted* attacks instead of the targeted attacks following the suggestions in [11] (a defense robust only to targeted adversarial attacks is weaker than one robust to untargeted adversarial attacks). In order to be comparable to the original paper in AT and ALP, we mainly evaluate the model under the $L_\infty$ bounded attacks. We also evaluate the models under other norm-bounded attacks ($L_0, L_2$). The PGD and 20PGD in our Table 1 refer to the PGD attacks with the random restart of 1 time and 20 times, respectively. For black-box (BB) attacks, we use the transfer based method [26]. We set $\lambda = 0.5$ for ALP method as the original paper. All the other implementation details are discussed in the appendix.

| | Attacks (Steps) | Clean - | FGSM (1) | BIM (40) | C&W (40) | PGD (40) | PGD (100) | 20PGD (100) | MIM (200) | BB (100) |
|---|---|---|---|---|---|---|---|---|---|---|
| **MNIST** | | | | | | | | | | |
| Methods | UM | 99.20% | 34.48% | 0% | 0% | 0% | 0% | 0% | 0% | 81.81% |
| | AT | 99.24% | 97.31% | 95.95% | 96.66% | 96.58% | 94.82% | 93.87% | 95.47% | 96.67% |
| | ALP | 98.91% | 97.34% | 96.00% | 96.50% | 96.62% | 95.06% | 94.93% | 95.41% | 96.95% |
| | TLA-RN | 99.50% | 98.12% | 97.17% | 97.17% | 97.64% | **97.07%** | 96.73% | **96.84%** | 97.69% |
| | TLA-SA | 99.44% | 98.14% | 97.08% | **97.45%** | 97.50% | 96.78% | 95.64% | 96.45% | 97.65% |
| | TLA | **99.52%** | **98.17%** | **97.32%** | 97.25% | **97.72%** | 96.96% | **96.79%** | 96.64% | **97.73%** |

| | Attacks (Steps) | Clean - | FGSM (1) | BIM (7) | C&W (30) | PGD (7) | PGD (20) | 20PGD (20) | MIM (40) | BB (7) |
|---|---|---|---|---|---|---|---|---|---|---|
| **CIFAR-10** | | | | | | | | | | |
| Methods | UM | **95.01%** | 13.35% | 0% | 0% | 0% | 0% | 0% | 0% | 7.60% |
| | AT | 87.14% | 55.63% | 48.29% | 46.97% | 49.79% | 45.72% | 45.21% | 45.16% | 62.83% |
| | ALP | 89.79% | **60.29%** | 50.62% | 47.59% | 51.89% | 48.50% | 45.98% | 45.97% | 67.27% |
| | TLA-RN | 81.02% | 55.41% | 51.44% | 49.66% | 52.50% | 49.94% | 45.55% | 49.63% | 65.96% |
| | TLA-SA | 86.19% | 58.80% | 52.19% | 49.64% | 53.53% | 49.70% | 49.15% | 49.29% | 61.67% |
| | TLA | 86.21% | 58.88% | **52.60%** | 50.69% | **53.87%** | 51.59% | **50.03%** | 50.09% | **70.63%** |

| | Attacks (Steps) | Clean - | FGSM (1) | BIM (10) | C&W (10) | PGD (10) | PGD (20) | 20PGD (20) | MIM (40) | BB (10) |
|---|---|---|---|---|---|---|---|---|---|---|
| **Tiny ImageNet** | | | | | | | | | | |
| Methods | UM | **60.64%** | 1.15% | 0.01% | 0.01% | 0.01% | 0% | 0% | 0% | 9.99% |
| | AT | 44.77% | 21.99 | 19.59% | 17.34% | 19.79% | 19.44% | 19.25% | 19.28% | 27.73% |
| | ALP | 41.53% | 21.53% | 20.03% | 16.80% | 20.18% | 19.96% | 19.76% | 19.85% | **30.31%** |
| | TLA-RN | 42.11% | 21.47% | 20.03% | 17.00% | 20.05% | 19.93% | 19.81% | 19.91% | 30.18% |
| | TLA-SA | 41.43% | 22.09% | 20.77% | 17.28% | 20.82% | 20.63% | **20.50%** | 20.61% | 29.96% |
| | TLA | 40.89% | **22.12%** | 20.77% | **17.48%** | 20.89% | 20.71% | 20.47% | **20.69%** | 29.98% |

Table 1: Classification accuracy under 8 different $L_\infty$ bounded *untargeted* attacks on MNIST ($L_\infty$ =0.3), CIFAR-10 ($L_\infty$ =8/255), and Tiny-ImageNet ($L_\infty$ =8/255). The best results of each column are in **bold** and the empirical lower bound (the lowest accuracy of each row if any) for each method is underlined. TLA improves the adversarial accuracy by up to 1.86%, 4.12% , and 0.84% on MNIST, CIFAR-10, and Tiny ImageNet respectively.

## 5.1 Effect of TLA on Robust Accuracy

**MNIST** consists of a training set of 55,000 images (excluding the 5000 images for validation as in [19]) and a testing set of 10,000 images. We use a variant of LeNet CNN architecture which has batch normalization for all the methods. The details of network architectures and hyper-parameters are summarized in the appendix. We adopt the $L_\infty = 0.3$ bounded attack during the training and evaluation. We generate adversarial examples using PGD with 0.01 step size for 40 steps during the training. In addition, we conduct different types of $L_\infty = 0.3$ bounded attacks to achieve good evaluations. The adversarial classification accuracy of different models under various adversarial attacks is shown in Table 1. As shown, we improve the empirical state-of-the-art adversarial accuracy by up to **1.86%** on 20PGD attacks (100 steps PGD attacks with 20 times of random restart), along with **0.28%** improvement on the clean data.

**CIFAR-10** consists of 32×32×3 color images in 10 classes, with 50k images for training and 10k images for testing. We follow the same wide residual network architecture and the same hyper-parameters settings as AT [19]. As shown in Table 1, our method achieves up to **4.12%** adversarial accuracy improvement over the baseline methods under the strongest 20PGD attacks (20 steps PGD attack with 20 times of restart). Note that our method results in a minor decrease of standard accuracy, but such loss of generic accuracy is observed in all the existing robust training models [38, 49]. The comparison with TLA-RN illustrates the effectiveness of the negative sampling strategy. According to the result of the TLA-SA, our selection of the adversarial example as the anchor also achieves better performance than the method which chooses the clean image as the anchor.

**Tiny Imagenet** is a tiny version of ImageNet consisting of color images with size 64×64×3 belonging to 200 classes. Each class has 500 training images and 50 validation images. Due to the GPU limit, we adapt the ResNet 50 architectures for the experiment. We adopt $L_\infty = 8/255$ for both training and validation. During training, we use 7 step PGD attack with step size $2/255$ to generate the adversarial samples. As shown in Table 1, our proposed model achieves higher adversarial accuracy under white box adversarial attacks by up to **0.84%** on MIM attacks.

| | | Mini-Batch TT (s) | Total TT (s) | Clean | FGSM(1) | BIM(7) | C&W(30) | PGD(20) | MIM(40) |
|---|---|---|---|---|---|---|---|---|---|
| Negative Size | 1 | 0 | 1.802 | 81.02% | 55.41% | 51.44% | 49.66% | 49.94% | 49.63% |
| | 250 | 0.467 | 2.259 | 86.38% | 59.05% | 53.02% | 50.49% | 50.71% | 50.31% |
| | 500 | 0.908 | 2.688 | **88.32%** | **60.02%** | 53.20% | **51.30%** | 50.46% | 50.07% |
| | 1000 | 1.832 | 3.621 | 86.71% | 59.08% | **53.25%** | 50.88% | **51.22%** | **50.74%** |
| | 2000 | 3.548 | 5.992 | 87.45% | 59.23% | 52.52% | 50.57% | 50.20% | 49.79% |

Table 2: The effect of mini-batch size of *negative* samples on training time (TT) per iteration and adversarial robustness ($L_\infty = 8/255$) on CIFAR-10 dataset. The best results of each column are shown in **bold**. The number of steps for each attack is shown in the parenthesis. The training time grows linearly as the size of the mini-batch grows. The adversarial robustness peaks at size 500 to 1000, which validate that semi-hard negative examples are crucial for TLA.

| | | MNIST (LeNet) | | | | CIFAR-10 (WRN) | | | |
|---|---|---|---|---|---|---|---|---|---|
| | Attacks | JSMA ($L_0$) | PGD ($L_2$) | C&W ($L_2$) | DeeoFool ($L_2$) | JSMA($L_0$) | PGD ($L_2$) | C&W ($L_2$) | DeeoFool ($L_2$) |
| Methods | AT | 99.08% | 96.61% | 99.08% | 99.13% | 40.4% | 36.8% | 50.0% | 67.7% |
| | ALP | 98.83% | 96.28% | 98.91% | 98.95% | 36.9% | 38.6% | 51.2% | 43.5% |
| | TLA | **99.32%** | **97.38%** | **99.36%** | **99.35%** | **48.6%** | **41.1%** | **53.5%** | **80.8%** |

Table 3: Classification accuracy of two baseline methods and TLA method on 4 unseen types of attacks ($L_0$ and $L_2$ norm bounded). All the models are only trained on the $L_\infty$ bounded attacks. The best results of each column are shown in **bold**. TLA improves the adversarial accuracy by up to 1.10% and 13.1% on MNIST and CIFAR-10 dataset respectively. The results demonstrate that TLA generalizes better to unseen types of attacks.

**Effect of the mini-batch size of negative samples of TLA.** Compared with retrieving from the whole dataset, the mini-batch based method can mitigate the computational overhead by finding the nearest neighbor from a batch rather than from the whole training set. The size of the mini-batch size controls the hardness level of the negative samples, where larger mini-batch size makes harder negative ones. We train models with different mini-batch size and evaluate the robustness of the model using five untargeted, $L_\infty$ bounded attacks. As shown in Table 2, the total training time grows linearly as the size of the mini-batch increases, which triples for size 2000 compared with size 1. The adversarial robustness first increases and then decreases after the mini-batch size reaches 1000 (very hard negative examples hurt performance). Being consistent with the observation in standard metric learning [32, 51], our results show that it is important to train TLA with semi-hard negative examples by choosing the proper mini-batch size.

**Generalization to Unseen Types of Attacks.** We evaluate the $L_\infty$ robustly trained models on unseen $L_0$-bounded [27] and $L_2$-bounded attacks [23, 5, 19, 5]. We set $L_0 = 0.1$ and $L_0 = 0.02$ bound for JSMA on MNIST and CIFAR-10 dataset respectively. For $L_2$ norm bounded PGD and C&W attacks, we set the bound as $L_2 = 3$ and $L_2 = 128$ on MNIST and CIFAR-10 respectively. We apply 40 steps of PGD and C&W on MNIST with step size 0.1, and 10 steps of PGD and C&W on CIFAR-10 with step size 32. We apply 2 steps for DeepFool attack for both dataset. Due to the slow speed of JSMA, we only run 1000 test samples on CIFAR-10. Table 3 shows that TLA improves the adversarial accuracy by up to 1.10% and 13.1% on MNIST and CIFAR-10 respectively, which demonstrates that TLA generalizes better to unseen attacks than baseline models.

**Performance on Different Model Architectures.** To demonstrate that TLA is general for different model architectures, we conduct experiments using multi-layer perceptron (MLP) and ConvNet [47] architectures. Results in Table 4 show that TLA achieves better adversarial robustness by up to 4.27% and 0.55% on MNIST and CIFAR-10 respectively.

| | | MNIST (MLP) | | | | | Cifar10 (ConvNet) | | | | |
|---|---|---|---|---|---|---|---|---|---|---|---|
| | Attacks | Clean | FGSM | BIM | C&W | PGD | Clean | FGSM | BIM | C&W | PGD |
| | Steps | - | 1 | 40 | 40 | 100 | - | 1 | 7 | 30 | 20 |
| Methods | UM | **98.27%** | 5.23% | 0% | 0% | 0% | **77.84%** | 3.50% | 0.09% | 0.08% | 0.03% |
| | AT | 96.43% | 73.25% | 57.83% | 62.60% | 58.10% | 67.60% | 40.26% | 36.34% | 33.17% | 34.83% |
| | ALP | 95.56% | 77.08% | 64.39% | 63.46% | 64.13% | 66.18% | 39.45% | 36.15% | 32.55% | 35.32% |
| | TLA | 97.15% | **78.44%** | **65.47%** | **67.73%** | **65.88%** | 67.48% | **40.76%** | **36.77%** | 33.27% | **35.38%** |

Table 4: Effect of TLA on different neural network architectures. The table lists classification accuracy under various $L_\infty$ bounded *untargeted* attacks on MNIST ($L_\infty = 0.3$) and Cifar10 ($L_\infty = 8/255$). Overall, TLA improves adversarial accuracy.

## 5.2 Effect of TLA on Adversarial vs. Natural Image Separation

Recall in Figure 2b and Figure 2c, the representations of adversarial images are shifted toward the false class. A robust model should separate them apart. To quantitatively evaluate how well TLA training helps with separating the adversarial examples from the natural images of the corresponding 'false' classes, we define the following metric.

Let $\{c_k^i\}$ denote the *embedded representations* of all the natural images from class $c_k$, where $i = 1, \ldots, |c_k|$, and $|c_k|$ is the total number of images in class $c_k$. Then, the average pairwise within-class distance of these embedded images is: $\sigma_{c_k}^{ntrl} = \frac{2}{|c_k|(|c_k|-1)} \sum_{i=1}^{|c_k|-1} \sum_{j=i+1}^{|c_k|} D(c_k^i, c_k^j)$. Let $\{c_k'^q\}$ further denote embedded representations of all the adversarial examples that are misclassified to class $c_k$, where $q = 1, \ldots, |c_k'|$, and $|c_k'|$ is the total number of such examples. Note that, class $c_k$ is the 'false' class to those adversarial images. Then, the distance between an adversarial images $c_k'^i$ and a natural image $c_k^j$ is: $D(c_k'^i, c_k^j)$, and the average pair-wise distance between adversary image and natural images is: $\sigma_{c_k}'^{adv} = \frac{1}{|c_k'||c_k|} \sum_{i=1}^{|c_k'|} \sum_{j=1}^{|c_k|} D(c_k'^i, c_k^j)$. We then define the ratio $r_{c_k} = \frac{\sigma_{c_k}^{adv}}{\sigma_{c_k}^{ntrl}}$ as a metric to evaluate how close the adversarial images are w.r.t. the natural images of the 'false' class while compared with the average pairwise within-class distance of all the natural images of that class. Finally, for all classes we compute the average ratio as $r = \frac{1}{M} \sum_{k=1}^{M} (r_{c_k})$. Note that, any good robust method should increase the value of $r$, indicating $\sigma^{adv}$ is far from $\sigma^{ntrl}$, i.e., they are better separated than the natural cluster, as shown in Figure 2d.

| Dataset | MNIST | | CIFAR-10 | | Tiny ImageNet | |
|---|---|---|---|---|---|---|
| Perturbation Level | $L_\infty = 0.03$ | $L_\infty = 0.3$ | $L_\infty = \frac{8}{255}$ | $L_\infty = \frac{25}{255}$ | $L_\infty = \frac{8}{255}$ | $L_\infty = \frac{25}{255}$ |
| AT | 1.288 | 1.308 | 1.053 | 1.007 | **0.9949** | 0.9656 |
| ALP | 1.398 | 1.394 | 1.038 | 1.210 | 0.9905 | 0.9722 |
| TLA | **1.810** | **1.847** | **1.093** | **1.390** | 0.9937 | **0.9724** |

Table 5: Average Ratio ($r$) of mean distance between adversary points and natural points over the mean intra-class distance. The best results of each column are in **bold**. The results illustrate that TLA increases the relative distance of adversarial images w.r.t. the natural images of the respective false classes, which illustrates that TLA achieves more desirable geometric feature space under attacks.

For every dataset, we estimate the ratios under two different perturbation levels of PGD attacks for all the models. As shown in Table 5, stronger attacks (larger perturbation level) tend to shift their latent representation more toward the false class. For Tiny-ImageNet, the adversarial examples are even closer ($r < 1$) to the false class's manifold than the corresponding natural images to themselves, which explains the low adversarial accuracy on this dataset. In almost all the settings, TLA leads to higher $r$ values of separation than the other baseline methods. This indicates TLA is most effective in pulling apart the misclassified adversary examples from their false class under both small and large perturbations attacks.

| Dataset | MNIST | | CIFAR-10 | | Tiny-ImageNet | |
|---|---|---|---|---|---|---|
| Type | Adv | Natural | Adv | Natural | Adv | Natural |
| AT | 93.01% | 98.68% | 47.46% | 87.06% | 20.20% | **36.6%** |
| ALP | 95.20% | 98.43% | 48.85% | **89.63%** | 20.33% | 35.23% |
| TLA | **96.98%** | **99.47%** | **51.74%** | 86.29% | **20.72%** | 33.99% |

Table 6: Accuracy of K-Nearest Neighbors classifier with $K = 50$, illustrating TLA has better similarity measures in embedding space even with adversarial samples. The best results of each column are in **bold**.

We further conduct the nearest neighbor analysis on the latent representations across all the models. The results illustrate the advantage of our learned representations for retrieving the nearest neighbor under adversarial attacks (See Figure 4). Table 6 numerically shows that the latent representation of TLA achieves higher accuracy using K-Nearest Neighbors classifier than baseline methods.

## 5.3 Effect of TLA on Adversarial Image Detection

Detecting mis-predicted adversarial inputs is another dimension to improve a model's robustness. Forward these detected adversarial examples to humans for labeling can significantly improve the reliability of the system under adversarial cases. Given that TLA separates further the adversarial examples from the natural examples of the false class, we can detect more mis-classified examples by filtering out the outliers. We conduct the following experiments.

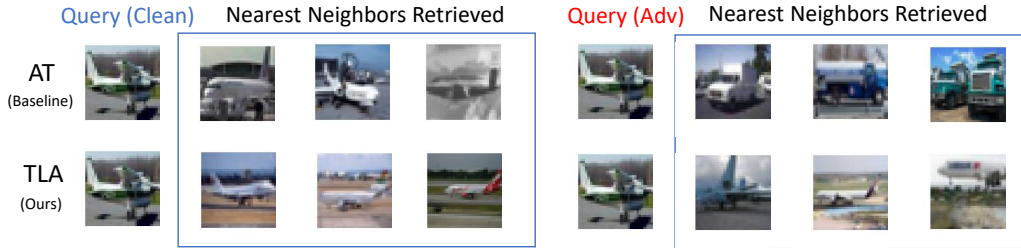

Figure 4: Visualization of nearest neighbor images while querying about a "plane" on AT and TLA trained models. For a natural query image, both methods retrieve correct images (left column). However, given an adversarial query image (right column), the AT retrieves false "truck" images indicating the perturbation moves the representation of the "plane" into the neighbors of "truck," while TLA still retrieves images from the true "plane" class.

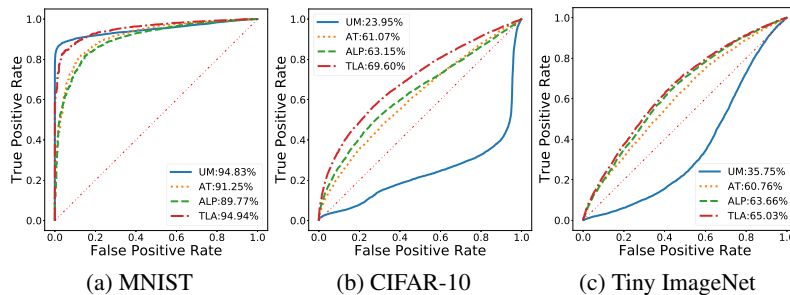

(a) MNIST          (b) CIFAR-10          (c) Tiny ImageNet

Figure 5: The ROC curve and AUC scores of detecting mis-classified adversarial examples. We train a GMM model on half clean and half adversarial examples (generated with perturbation level $\epsilon = 0.03/1(40 \text{ steps})$ for MNIST, $\epsilon = 8/255(7 \text{ steps})$ for CIFAR-10, and $\epsilon = 8/255(7 \text{ steps})$ for Tiny-ImageNet), and then test the detection model on 10k natural test images and 10k adversary test images (generated with perturbation level $\epsilon = 0.3/1(100 \text{ steps})$ for MNIST, $\epsilon = 25/255(20 \text{ steps})$ for CIFAR-10, and $\epsilon = 25/255(30 \text{ steps})$ for Tiny-ImageNet). The numerical results for AUC score are shown in the legend. Note that both the ROC curve of TLA is on the top and the AUC score of TLA is the highest, which shows TLA (our method) achieves higher detection efficiency for adversarial examples.

Following the adversarial detection method proposed in [52], we train a Gaussian Mixture Model for 10 classes where the density function of each class is captured by one Gaussian distribution. For each test image, we assign a confidence score of a class based on the Gaussian distribution density of the class at that image, as shown in [45]. We assign these confidence scores for all the 10 classes for each test image. We then pick the class with the largest confidence value as the assigned class of the image. We further rank all the test images based on the confidence value of their assigned class. We reject those with lower confidence scores below a certain threshold. This method serves as an additional confidence metric to detect adversarial examples in a real-world setting.

We conduct the detection experiment for mis-classified images on 10k clean images and 10k adversarial images. As shown in Figure 5, the ROC-curves and AUC score demonstrate that our learned representations are superior in adversarial example detection. Compared with other robust training models, TLA improves the AUC score by up to 3.69%, 6.45%, and 1.37% on MNIST, CIFAR-10, and Tiny ImageNet respectively. The detection results here are consistent with the visual results shown in Figure 2.

# 6 Conclusion

Our novel TLA regularization is the first method that leverages metric learning for adversarial robustness on deep networks, which significantly increases the model robustness and detection efficiency. TLA is inspired by the evidence that the model has distorted feature space under adversarial attacks. In the future, we plan to enhance TLA using more powerful metric learning methods, such as the N-pair loss. We believe TLA will also be beneficial for other deep network applications that desire a better geometric relationship in hidden representations.

## 7 Acknowledgements

We thank the anonymous reviewers, Prof. Suman Jana, Prof. Shih-Fu Chang, Prof. Janet Kayfetz, Ji Xu, Hao Wang, and Vaggelis Atlidakis for their valuable comments, which substantially improved our paper. This work is in part supported by NSF grant CNS-15-64055; ONR grants N00014-16-1-2263 and N00014-17-1-2788; a JP Morgan Faculty Research Award; a DiDi Faculty Research Award; a Google Cloud grant; an Amazon Web Services grant; NSF CRII 1850069; and NSF CCF-1845893.

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
