[Supplementary Material]

# Supplementary material for "Metric Learning for Adversarial Robustness"

## A    Indiscrimination of Robust Representation of Adversarial Examples and the True Class

Similar to Fig. 1 in the main text, we visualize the representations of clean and adversarial examples from the same class for the remaining 9 classes on CIFAR-10 dataset across all models using t-SNE [1]. Visualizations are shown in Fig. 1, Fig. 2, and Fig. 3.

Figure 1:  **t-SNE Visualizations of adversarial images from the same *true* class which are mistakenly classified to *false* classes. From left to right: UM, AT, ALP, TLA.** These are representations of second to last layer of 1000 adversarial examples crafted from 1000 clean test examples from CIFAR-10 dataset, where the *true* class is the same for all the figures in the same row and different for figures of different row. The different colors represent different *false* classes. The gray dots further show 500 randomly sampled clean *true* images. Notice that for (a) undefended model (UM), the adversarial attacks clearly separate the images from the *true* category into different classes. (b) adversarial training (AT) and (c) adversarial logit pairing (ALP) method still suffer from this problem at a reduced level. In contrast, proposed ATL (see (d)) clusters together all the examples from the same *true* class, which improves overall robustness.

## B    Separation of Robust Representations of Adversarial Examples to the False Class

Similar to Fig. 2 in the main text, we provide more visualizations of the representations on CIFAR-10 using t-SNE to demonstrate the separation margin of adversarial samples to the corresponding false class. We plot the representations of adversarial examples from class $A$ which are finally misclassified as class $B$. We also plot clean images from both $A$ and $B$. Visualizations are shown in Fig. 4, Fig. 5, and Fig. 6.

Figure 2: **t-SNE Visualizations of adversarial images from the same *true* class which are mistakenly classified to different *false* classes (Same as Fig 1).**

# C   Experiment

## C.1   Implementation Details

We add uniform random noise to the clean images $\mathbf{x}$ within the $\mathbf{I}(\mathbf{x}, \epsilon)$ corresponding to the allowed perturbation scale for the natural images in the TLA training. We conduct all the experiments using a single V100 GPU with 16GB memory. The black-box attack is evaluated Code is appended in the zip file.

**MNIST** follows the setup of Madry et al. [4] and ALP [3], we use Adam with learning rate of 0.0001. For ALP, we use $\lambda = 0.5$ as suggested in papers [3]. We conduct experiments using our modified LeNet model (adding the batch normalization and replace $5 \times 5$ convolution kernel with $3 \times 3$). The architecture is shown in Table 1. All experiments are conducted with batch size 50. To be consistent with the results reported by ALP, we maintain the label smoothing with value equals to 0.1. We achieve better accuracy for AT [4] and ALP [3] (The baselines are stronger than the original paper because of the additional label smoothing and batch normalization). We set up the experiment we reported in the table with the following hyper-parameters. For the TLA method, we adopt $\lambda_1 = 0.5$, $\lambda_2 = 0.001$, margin $\alpha = 0.05$, mini-batch size for the negative sample selection as 50. We run the experiments for 200 epochs before it fully converges. We repeat the experiments for five times and observe little oscillations for the performance. We select the one randomly with middle-level performance and conduct all the evaluations above.

**CIFAR10** follows the same WRN model as Madry et al [4] across all our models, as shown in Table 2. Also, we adopt the same SGD optimization method with the same learning rate decay strategy as Madry's, where we start with learning rate of 0.1 and decrease it to 0.01 at 50k iterations. We run it for 55k iterations before stopping. We train all the models with a batch size of 50. We implement the ALP on CIFAR-10 because it is not implemented in the original ALP paper, where we do improve the adversarial accuracy significantly. To achieve a fair comparison, we all follow the hyper-parameters

Figure 3: **t-SNE Visualizations of adversarial images from the same *true* class which are mistakenly classified to different *false* classes (Same as Fig 1).**

Figure 4: **Illustration of the separation margin of adversarial examples from the natural images of the corresponding false class. From left to right: UM, AT, ALP, TLA.** We show t-SNE visualization of the second to last layer representation of test data from two different classes across four models. The blue and green dots are 200 randomly sampled natural images from "frog" and "horse" classes respectively. The red triangles denote adversarial (adv) perturbed "frog" images but mispredicted as "horse". Notice that for (a) UM, the adversarial examples are moved to the center of the false class which is hard to separate from the natural images of the false class. (b) AT and (c) ALP achieve some robustness by separating adversarial and false natural images, but they are still close to each other. Plot (d) shows proposed TLA promotes the mispredicted adversarial examples to lie on edge and can still be separated from natural images of the false class, which improves the robustness.

set-up in [4]. It took a day and a half before our training converges. We set the $\lambda = 0.5$ for ALP and do not use label smoothing. For TLA method, we adopt $\lambda_1 = 2$, $\lambda_2 = 0.001$, margin $\alpha = 0.03$, mini-batch size for the negative sample selection as 500. Our TLA improves the robust accuracy over ALP baseline for **4.12%** and AT baseline for **4.82%**.

**Tiny-ImageNet** follows the well studied Resnet-50 model. We apply a stride of 2 for the first convolution to reduce the computational intensity. We use the Adam optimizer for AT and ALP trained with batch size of 32. We start from learning rate of 0.1 and decrease the learning rate to 0.01 at the 110-th epochs and 0.001 at the 130th epochs. We train it for 150 epochs. For TLA

Figure 5: **Same as Fig 4 except the two classes are "horse" and "airplane".**

Figure 6: **Same as Fig 4 except the two classes are "horse" and "airplane".**

training, we fintuning on the AT model with batch-size of 20 because of GPU memory budget. We use the untargeted attacks for the adversarial examples generation during both the training and testing procedure, so that it is consistent with the attack conducted in the evaluation. We use data augmentation (crop, flip, saturation, etc.) for all the models. The training time for the models requires about 2 days on a single Nvidia V100 GPU. We set the $\lambda = 0.5$ for ALP. We use label smoothing with parameter equal to 0.1 across all of our experiments. For the TLA method, we adopt $\lambda_1 = 0.2$, $\lambda_2 = 0.001$, margin $\alpha = 0.01$, mini-batch size for the negative sample selection as 50.

Table 1: Illustration of MNIST architecture, which shows all the details of our modified LeNet Architecture by using smaller Convolution (Conv) kernels and batch normalization. Where the Feature-In/Out for the convolution and fully connected (FC) denotes the number of the channel and hidden neurons respectively.

| Layer Type | Feature-In Dimension | Feature-Out Dimension | Kernel-Size |
|---|---|---|---|
| Conv | 1 | 32 | $3 \times 3$ |
| BatchNormalization | 32 | 32 | - |
| ReLU | - | - | - |
| Conv | 32 | 64 | $3 \times 3$ |
| BatchNormalization | 64 | 64 | - |
| ReLU | - | - | - |
| Max Pooling $2 \times 2$ | | | |
| Conv | 64 | 128 | $3 \times 3$ |
| BatchNormalization | 128 | 128 | - |
| ReLU | - | - | - |
| Conv | 128 | 256 | $3 \times 3$ |
| BatchNormalization | 256 | 256 | - |
| ReLU | - | - | - |
| Max Pooling $2 \times 2$ | | | |
| Fully Connected | $7 \times 7 \times 256$ | 1024 | - |
| BatchNormalization | 1024 | 1024 | - |
| ReLU | - | - | - |
| Fully Connected | 1024 | 10 | - |
| Softmax | | | |

Table 2: Illustration of wide residual network architecture which is the same as Madry et al. [4]. The matrix denotes the set up of the residual block, and × denotes to repeat this for the following number of times.

| layer name | output size | layer setup | |
|---|---|---|---|
| Conv | $32 \times 32$ | $3 \times 3$, 16, stride 1 | |
| conv2_x | $16 \times 16$ | $3 \times 3, 160$ <br> $3 \times 3, 160$ | $\times 6$ |
| conv3_x | $8 \times 8$ | $3 \times 3, 320$ <br> $3 \times 3, 320$ | $\times 6$ |
| conv4_x | $4 \times 4$ | $3 \times 3, 640$ <br> $3 \times 3, 640$ | $\times 6$ |
| classifier | $1 \times 1$ | average pool, 10-d fc, softmax | |

## C.2 Effect of TLA on Bring Adversarial vs. Natural Image of the Same Class Together

We also define a similar metric to show TLA tends to pull closer the adversary images to their true class. For every dataset, we compute a complementary ratio (denoted as $r'$) that measures how adversary images are pulled back to their true class on different models. We reformulate the definition of $\{c_k'^q\}$ to be the embedded representations of all the adversarial examples crafted based on the clean images of true class $c_k$ in the test set. The results are shown in Table 3. Notice that lower value of $r'$ is desirable here, indicating the examples of the same class are pulled together.

As we can see, adversarial attack tends to bring the representation of an image far away from its true class. For UM, the adversarial examples are far away from the clean examples of the same class. With AT and ALP (baseline) methods, the adversarial examples are getting closer to the clean images of the true class to some extent. Our method TLA brings even closer the adversarial examples to the clean examples on CIFAR-10 and Tiny-ImageNet and achieves comparable performance on MNIST. This further implies that our method promotes the adversarial and clean images from the same class to lie on the same manifold and thus improves the robustness of the model.

Table 3: Average (over all classes) ratio ($r'$) of the mean of pairwise distance between adversary images and natural images of the same class over the mean inner-class distance. The results illustrate that TLA decreases the relative distance of adversarial images w.r.t. the natural images of the respective true classes. The best results of each column are in **bold**.

| | Dataset | MNIST | | CIFAR-10 | | Tiny-ImageNet | |
|---|---|---|---|---|---|---|---|
| | Perturbation Level | $L_\infty = 0.03$ | $L_\infty = 0.3$ | $L_\infty = \frac{8}{255}$ | $L_\infty = \frac{25}{255}$ | $L_\infty = \frac{8}{255}$ | $L_\infty = \frac{25}{255}$ |
| Methods | UM | 1.071 | 2.159 | 3.604 | 3.682 | 1.319 | 1.480 |
| | AT | **1.004** | **1.042** | 1.342 | 1.714 | 1.053 | 1.204 |
| | ALP | 1.006 | 1.068 | 2.313 | 3.796 | **1.040** | **1.151** |
| | TLA | 1.005 | 1.072 | **1.191** | **1.491** | 1.044 | 1.174 |

## D TLA Algorithm

The Triplet Loss Adversarial Training (TLA) is introduced in the Algorithm 1. It is a simple approach which can be done within one Loop.

## E The effect of the hyper-parameter

We use MNIST dataset to explore the influence of the hyper-parameters. We conclude that a higher accuracy is usually achieved with a margin between 0.01 to 0.1, the weight $\lambda$ should be between 0.5 to 2. The results for different margin and $\lambda$ plot in the following graph. Overall, our TLA algorithm does not sensitive to the specific hyper-parameters set up. In a wide range, it is able to achieve significant improvement over the baseline models.

---
**Algorithm 1** Metric Learning for Adversarial Robustness (Triplet Loss Adversarial (TLA) method)
---
1: **Input:** Data $\mathcal{D} = \{(\mathbf{x}^{(i)}, y^{(i)})\}_{i=1}^N$, training iterations $T_t$, learning rate $\rho_t$, initialized trainable model parameters $\boldsymbol{\theta}$. A minibatch of size $K$ for each iteration is denoted as $\{(\mathbf{X}^{(k)}, Y^{(k)})\}_{k \in \{i_1, \ldots, i_K\}}$.

2: **for** $t = 1 : T_t$ **do**

3:    Sample a minibatch of data $\mathbf{X}$ and $\mathbf{X}_{pos}$ of the same class from $\mathcal{D}$

4:    Generate adversarial attack images $\mathbf{X}_{adv}$ based on $\mathbf{X}$.

5:    Sample a subset of data $\mathbf{X}_{extra}$ and calculate a negative minibatch $\mathbf{X}_{neg}^-$ corresponding to $\mathbf{X}_{adv}$ with strategy mentioned in section 3.2.

6:    Calculate $\mathcal{L}_{all}$ (as defined in Sec 4.2) on the sampled batches.

7:    Update parameters: $\boldsymbol{\theta} \leftarrow \boldsymbol{\theta} - \rho_t \sum_k \nabla_{\boldsymbol{\theta}} \mathcal{L}_{all}$.

8: **end for**
---

Table 4: Adversarial accuracy under 100 steps of PGD attack when model is trained using different $\lambda_1$ parameters on MNIST. We conclude that setting the $\lambda_1$ within range of 0.5 to 2 is all reasonable.

| $\lambda$ | 0 | 0.025 | 0.5 | 1 | 2 | 4 |
|---|---|---|---|---|---|---|
| TLA | 94.82% | 96.31% | **96.96%** | 96.57% | 96.72% | 96.26% |

# F  Visualization

## F.1  More Visualization of the Nearest Neighbor Retrival on Learned Embeddings

We show more visualizations of the nearest neighbor retrieval based on the representation learned on different methods. The results are shown in Fig 7.

## F.2  Visualization of the Loss Landscape

To demonstrate that our approach does not rely on the obfuscated gradients by having a distorted loss landscape [2], we visualize the loss landscape of the loss function on two random directions in Fig 8.

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

Table 5: Adversarial accuracy under 100 steps of PGD attack with $\lambda_1 = 2$ when model is trained using different $\alpha$ parameter on MNIST. The best accuracy is achieved with margin 0.05 according to our experiment.

| $\alpha$ | 0 | 0.025 | 0.05 | 0.1 | 0.2 |
|---|---|---|---|---|---|
| TLA | 96.60% | 96.47% | **96.72%** | 96.36% | 96.35% |

Table 6: Adversarial robustness accuracy under 100 steps of PGD of model trained on different representation layers with ATL on MNIST. All the models using $\lambda_1 = 2$. The result demonstrate our choice of the second to last layer achieve the best performance.

| Representation Layer | Lower | Middle | Higher (Ours) | Logit (ALP) |
|---|---|---|---|---|
| TLA | 96.14% | 96.48% | **96.72%** | 96.34% |

Figure 7: **Visualization of nearest neighbor images while querying about a "deer" on models using AT, ALP, and TLA training separately.** The clean image query is shown on the left column, and the adversarial perturbed image query is shown on the right column. As we can see, while both baseline methods are unable to retrieve the correct nearest neighbors under adversarial attacks, our TLA method (bottom) retrieve the correct images.

Figure 8: **Visualization of loss landscape of each model of Tiny ImageNet.** We visualize the loss using heatmap of three randomly sampled example (each column has the same direction). For each line, we show the result of baseline methods and our methods. As we can see, TLA (last row) has a slightly smoother loss landscape compared with AT and ALP baselines.