[Reviews · NeurIPS 2019]

Reviewer 1



The paper proposes the use of triplet loss in order to achieve more desirable geometric relationships between an example, its adversarial counterpart and examples from the other classes. This is an intuitive proposal and the triplet loss has proven very useful in other ML contexts. The paper is generally well written and the triplet loss proposal for adversarial examples is original, to the best of my knowledge. Experimental results demonstrate the use of the proposed loss is very promising for improving adversarial robustness. It doesn't seem at all suprising that the latent representation of an adversarial example is shifted towards the false class (or away from the true class). After all, isn't this the basis of the optimization used to generate adversarial examples in the first place? Furthermore, using t-SNE to visualize the behaviour of adversarial examples is not a new idea and has been used in a variety of papers in a similar way, see e.g. -Generalizability vs. Robustness: Adversarial Examples for Medical Imaging -Defend Deep Neural Networks Against Adversarial Examples via Fixed andDynamic Quantized Activation Functions -IMPROVING THE GENERALIZATION OF ADVERSARIAL TRAINING WITH DOMAIN ADAPTATION For these reasons, I do not believe this component of the contribution is as significant as the proposal of the new loss function. A question for the future would the underlying idea be further refined to the popular quadruplet loss, which is often used to extend the triplet loss.

Reviewer 2



This paper analyzes the property of high dimensional latent representation of adversarial examples, finding that the attack makes the embedding move closer to the false class so that the adversarial and natural images are almost indistinguishable. The authors implemented the proposed method on several datasets including MNIST, CIFAR-10, Tiny Imagenet and achieved good performance. They also made comparison with different adversarial methods including FGSM, BIM, C&W, PGD and so on. This paper was in general well written. The authors provided a lot of visualization about their analysis and result. The authors also uploaded their code in the supplementary and the experiments seems sufficient and support strongly for their views. Some major concerns are listed as follows: 1. As a general framework, the authors implemented their idea on only one model structure. This seems insufficient. More experiments on a different model structure may make the paper more convincing. 2. Although, the paper provided a lot of experiments and visualization, the authors analyze adversarial examples only from the experiments, more theoretical analysis may be needed.

Reviewer 3



a) Originality:This paper propose a new method based on metric learning, and related work cited adequately. b) Quality: The paper is well written in general. It has the intuitive explanation for the motivation, the theoretical analysis and the experiment results. And experimental results support the authors' claim. c) Clarity:The paper is written clearly and easy to understand. d) Significance: The paper provides a new method which is easy to understand and implement. e) Some concerns: 1) Differ from other works' visualization of the last layer, this paper shows that of latent representation. Authors have said that the penultimate layer tends to have more information, but the proposed triplet loss is added on the output layer, is there a gap between the visualization and the proposed algorithm TLA? 1) Why do authors focus on the infinite norm adversarial attack? Is it the limitation of the proposed method? Can the method be applied to other ones? 2) It is too few to choose only AT and ALP as baseline. 3) There is a mistake in Fig 3. The figure does not agree with the notations in this paper.

[Author Response · NeurIPS 2019]

We thank the reviewers [R1, R2, R3] for their valuable and constructive comments. We are glad that all of them
appreciate the novelty of the proposed metric learning approach for adversarial robustness. Below we address the
reviewers' comments in detail.

**The Novelty of t-SNE Visualization of Latent Representation of Adversarial Examples [R1].** The main contribu-
tion of our paper is a triplet loss function for adversarial robustness, not t-SNE. We only use t-SNE visualization to
analyze and motivate a triplet loss function for adversarial robustness. We will follow R1's suggestion to clarify this
point in our paper and cite and discuss all related papers suggested by R1.

**Variants of Metric Learning [R1].** Thank you for the suggestion. Our approach is general and will work with several
metric learning approaches. We will mention these.

**Theoretical Analysis [R2].** We agree with R2 that theoretical understanding of adversarial samples for Neural Network
is important. Our paper presents an empirical visualization of robust vs. non-robust NNs under adversarial attacks and
uses it to motivate the proposed metric learning approach for adversarial robustness. Similar to many prior problems,
empirical work sometimes precedes theory [2][4]. In this paper, we focus on an empirical understanding, and theoretical
analysis is out of the scope.

**Variants of Model Architecture [R2, R3].** We present the same evaluation and architecture setup as our baseline
papers, where they also use one model architecture for each dataset. We evaluate different model architectures (LeNet,
WRN, ResNet50), which demonstrate that our TLA is not restricted to a single model architecture [R2]. Thank you
for the suggestion to compare different architectures, and we have included these results for MLP and ConvNet in the
following table. Overall, we observe similar improvements using our proposed TLA method.

| | | MNIST (MLP) | | | | | Cifar10 (ConvNet) | | | |
|---|---|---|---|---|---|---|---|---|---|---|
| | Attacks $L_\infty$ | Clean | FGSM | BIM | C&W | PGD | Clean | FGSM | BIM | C&W | PGD |
| Methods | UM | **98.27%** | 5.23% | 0% | 0% | 0% | **77.84%** | 3.50% | 0.09% | 0.08% | 0.03% |
| | AT | 96.43% | 73.25% | 57.83% | 62.60% | 58.10% | 66.06% | 40.20% | 36.32% | 34.34% | 34.80% |
| | ALP | 95.56% | 77.08% | 64.39% | 63.46% | 64.13% | 66.18% | 39.45% | 36.15% | 32.55% | 35.32% |
| | TLA | 97.15% | **78.44%** | **65.47%** | **67.73%** | **65.88%** | 66.53% | **41.03%** | **37.05%** | **34.50%** | **35.74%** |

**Gap between Visualization and the TLA layers [R3].** We apologize for the confusion. We indeed applied triplet loss
to the penultimate layer (line 137-138,174-179), i.e., the same layer we visualized using t-SNE. Thus, *no gap exists*
*between the visualization and the TLA layer*. We will fix this typo in the next revision.

**Baseline Defenses [R3].** We compare against state-of-the-art baselines for defending against $L_\infty$ attacks: (i) Adversarial
Training (AT) is an established model for adversarial robustness, and (ii) ALP achieved the state-of-the-art result on
adversarial robustness for ImageNet. We demonstrate that by adding a simple triplet loss during the adversarial training,
we substantially improve robustness over these state-of-the-art defenses. Other types of defense methods are either
orthogonal to our line of research or compatible with our defense method and can be applied simultaneously (e.g.
pre-training [2], ensemble learning [3], adding feature denoising blocks into model [4], leveraging additional unlabeled
data [1], etc.).

**Attack Types [R3].** An advantage of our approach is that it is general and works with several attack types. We originally
focused on infinite norm adversarial attack in order to fairly compare to baseline papers. However, as suggested by
R3, we also conducted evaluations on $L_0$ (JSMA) and $L_2$ (CW, PGD, DeepFool) attacks. The table below reports the
results, which follows the same trend as the original paper.

| | | MNIST (LeNet) | | | | Cifar10 (WRN) | | | |
|---|---|---|---|---|---|---|---|---|---|
| | Attacks | JSMA $(L_0)$ | PGD $(L_2)$ | CW $(L_2)$ | DeeoFool $(L_2)$ | JSMA$(L_0)$ | PGD $(L_2)$ | CW $(L_2)$ | DeeoFool $(L_2)$ |
| Methods | AT | 99.08% | 96.61% | 99.08% | 99.13% | 40.4% | 36.8% | 50.0% | 67.7% |
| | ALP | 98.83% | 96.28% | 98.91% | 98.95% | 36.9% | 38.6% | 51.2% | 43.5% |
| | TLA | **99.32%** | **97.38%** | **99.36%** | **99.35%** | **48.6%** | **41.1%** | **53.5%** | **80.8%** |

**Figure 3 [R3]** Thanks for pointing this out. There is a typo on line 136, where $\mathbf{x_a^{(i)}}$ should be $\mathbf{x_a'^{(i)}}$, which should be
consistent with Fig 3 and equation (1). We will fix this.

**References**

[1] Y. Carmon, A. Raghunathan, L. Schmidt, P. Liang, and J. C. Duchi. Unlabeled data improves adversarial robustness. *arXiv*
*preprint arXiv:1905.13736*, 2019.
[2] D. Hendrycks, K. Lee, and M. Mazeika. Using pre-training can improve model robustness and uncertainty. In *ICML*, 2019.
[3] T. Pang, K. Xu, C. Du, N. Chen, and J. Zhu. Improving adversarial robustness via promoting ensemble diversity. In *ICML*, 2019.
[4] C. Xie, Y. Wu, L. van der Maaten, A. L. Yuille, and K. He. Feature denoising for improving adversarial robustness. In *CVPR*,
2019.


[Meta-Review · NeurIPS 2019]

The paper presents a smart heuristic approach based on triplet loss to address adversarial attacks. The triplet loss considers, besides the current example, a nearest-neighbor (according to the latent representation) from another class as negative example, and another example of the same class as positive example. While the idea is intuitive, and nicely explained in the paper, the careful experimental validation shows it is empirically efficient on two datasets with respect to a number of attacks. The computational cost (finding the nearest negative) is mitigated by restricting the search to the current mini-batch. Please analyze the sensitivity and speed of convergence w.r.t. the minibatch size in the camera-ready.